# Multilevel Social Determinants of Patient-Reported Outcomes in Young Survivors of Childhood Cancer

**DOI:** 10.3390/cancers16091661

**Published:** 2024-04-25

**Authors:** Jin-ah Sim, Madeline R. Horan, Jaesung Choi, Deo Kumar Srivastava, Gregory T. Armstrong, Kirsten K. Ness, Melissa M. Hudson, I-Chan Huang

**Affiliations:** 1Department of Epidemiology & Cancer Control, St. Jude Children’s Research Hospital, Memphis, TN 38105, USA; jin-ah.sim@hallym.ac.kr (J.-a.S.); madeline.horan@stjude.org (M.R.H.); jaesung.choi@stjude.org (J.C.); greg.armstrong@stjude.org (G.T.A.); kiri.ness@stjude.org (K.K.N.); melissa.hudson@stjude.org (M.M.H.); 2Department of AI Convergence, Hallym University, Chuncheon 24252, Republic of Korea; 3Department of Biostatistics, St. Jude Children’s Research Hospital, Memphis, TN 38105, USA; kumar.srivastava@stjude.org; 4Department of Oncology, St. Jude Children’s Research Hospital, Memphis, TN 38105, USA

**Keywords:** patient-reported outcomes, pediatric cancer survivors, social determinants of health

## Abstract

**Simple Summary:**

This cross-sectional study analyzed how multilevel social factors affect patient-reported outcomes in children under 18 who survived cancer. Study participants include 293 pairs of these survivors who received survivorship care at a U.S.-based comprehensive cancer center between 2017 and 2018 and their primary caregivers. Findings indicate that higher caregiver anxiety is linked to worse depression, stress, fatigue, sleep problems, and lower positive affect in survivors of pediatric cancer. The study also found that family conflicts are associated with sleep issues in childhood cancer survivors. Furthermore, survivors living in socioeconomically deprived areas experience poorer sleep quality, and those residing in environments with high physical deprivation experience more psychological stress and fatigue, alongside reduced positivity and mobility. These results highlight the significant impact of parental, familial, and neighborhood factors on a range of patient-reported outcomes among young cancer survivors, suggesting these social factors as crucial targets for intervention.

**Abstract:**

In this study, the social determinants of patient-reported outcomes (PROs) in young survivors of childhood cancer aged <18 years are researched. This cross-sectional study investigated social determinants associated with poor PROs among young childhood cancer survivors. We included 293 dyads of survivors receiving treatment at St. Jude Children’s Research Hospital who were <18 years of age during follow-up from 2017 to 2018 and their primary caregivers. Social determinants included family factors (caregiver-reported PROs, family dynamics) and county-level deprivation (socioeconomic status, physical environment via the County Health Rankings & Roadmaps). PROMIS measures assessed survivors’ and caregivers’ PROs. General linear regression tested associations of social determinants with survivors’ PROs. We found that caregivers’ higher anxiety was significantly associated with survivors’ poorer depression, stress, fatigue, sleep issues, and reduced positive affect (*p* < 0.05); caregivers’ sleep disturbances were significantly associated with lower mobility in survivors (*p* < 0.05). Family conflicts were associated with survivors’ sleep problems (*p* < 0.05). Residing in socioeconomically deprived areas was significantly associated with survivors’ poorer sleep quality (*p* < 0.05), while higher physical environment deprivation was associated with survivors’ higher psychological stress and fatigue and lower positive affect and mobility (*p* < 0.05). Parental, family, and neighborhood factors are critical influences on young survivors’ quality of life and well-being and represent new intervention targets.

## 1. Introduction

Advances in cancer treatment and supportive care have significantly improved the survival rates of children with cancer such that >85% will become five-year survivors [1,2,3]. The notable increase in survivorship rates has prompted a shift in focus from solely emphasizing the duration of survival to understanding the impact of childhood cancer treatment on the quality of life and perceived well-being. This impact can be assessed through the use of patient-reported outcome (PRO) measures [4]. Due to their cancer treatment, survivors often develop long-term sequelae, including co-morbid physical and psychological conditions [5,6,7,8,9]. Numerous cancer survivors were initially diagnosed and received treatment during their early years of life (e.g., the typical age for a pediatric leukemia diagnosis ranges from 2 to 5 years). Consequently, the emergence of delayed complications can begin in late childhood and adolescence [10,11,12] and may continue into adulthood [13,14].

Psychosocial difficulties are not only present in survivors but are also reported by family members and caregivers of children who have survived cancer. Parental characteristics, including low educational attainment, parenting style, and poor health states, can significantly impact the health outcomes of pediatric cancer survivors [15,16,17]. Parental worry about survivorship and late effects and their depression and anxiety status further contribute to adverse health outcomes (e.g., poor health-related quality of life (HRQOL) and symptoms) in pediatric cancer survivors [16,17,18]. Family cohesion and conflict are recognized as key indicators of family dynamics, carrying significant implications for the social, emotional, and behavioral adjustment of adolescents. Previous studies suggest that low cohesion or high conflict within the families are likely associated with prevalent symptoms among pediatric cancer individuals [19,20].

In addition to their family environment, the outcomes for childhood cancer survivors are also influenced by their wider social and physical surroundings [21,22,23]. Per recommendations by the Children’s Oncology Group Long-Term Follow-Up Guidelines, maintaining a healthy lifestyle (e.g., physical activity, healthy diet) after cancer therapy is key for preventing late effects among survivors of childhood, adolescent, and young adult cancers [24]. However, children residing in underprivileged areas commonly experience a shortage of physical infrastructure and social resources, including poor-quality food, housing, transportation, and social networks, challenging adherence to healthy lifestyles [25,26]. A deprived environment may also trigger psychological stress, which in turn causes psychological, somatic, and physical symptoms and long-term disabilities [27].

Prior research in pediatric cancer survivorship has identified certain demographic (e.g., older age, female sex) and treatment (e.g., region of radiation, agent of chemotherapy) risk factors for incident chronic health conditions [28,29], poor PROs [30,31], and premature mortality [32,33]. However, these studies largely focus on long-term, aging adult survivors of childhood cancer rather than young survivors aged <18 years. Therefore, this study explored risk factors of physical and psychosocial PROs in cancer survivors aged 8–17.9 years, with a focus on multilevel social determinants at the family and neighborhood levels.

## 2. Methods

### 2.1. Study Participants and Data Collection

This cross-sectional study included 293 dyads of childhood cancer survivors and their primary caregivers who visited St. Jude Children’s Research Hospital (SJCRH) between July 2017 and June 2018 for annual follow-up. Inclusion criteria were children/adolescents aged 8–17.9 years at the time of assessment, a diagnosis of cancer/malignancy during childhood that was treated at SJCRH, and a period of at least 5 years from their initial cancer diagnosis. We excluded survivors with developmental delay/intellectual disability or lack of a complete home address for geocoding (Appendix A for flow diagram).

Survivors completed a survey of pediatric PROs, and primary caregivers completed a survey of the child and caregiver’s demographic information, caregiver’s SES (educational attainment, marital status, employment, household income, health insurance), their PROs and parenting behaviors, and family dynamics. We used a geocoding approach to obtain the Federal Information Processing Standards (FIPS) codes based on the home address information of each participant, including the structure number, street name, city, state, and zip code. For this purpose, we utilized the Geocoding Service API provided by the U.S. Census Bureau (https://geocoding.geo.census.gov/geocoder/ (accessed on 22 April 2024)) to transform each address into geographical coordinates, specifically latitude and longitude. Using the latitude and longitude data, we linked the address of each study participant to FIPS codes, which provide location-specific information in the U.S., including demographic, environmental, and administrative datasets. Detailed cancer diagnosis and treatment information (chemotherapy, radiotherapy, surgery) were abstracted from medical records.

This study was approved by the SJCRH Institutional Review Board, and survivors and caregivers completed informed assent and consent, respectively, before study participation.

### 2.2. Survivor and Caregiver PROs

Patient-Reported Outcomes Measurement Information System (PROMIS^®^) pediatric short forms assessed the survivor’s depression, psychological stress, fatigue, sleep disturbance, positive affect, and mobility domains. PROMIS^®^ adult short forms assessed the caregiver’s anxiety, depression, fatigue, sleep disturbance, pain severity, and physical function. Each PRO domain was scored on a T-metric (mean = 50, standard deviation [SD] = 10). For the positive affect and mobility domains in survivors and physical function in caregivers, higher scores represent better PROs; otherwise, higher scores reflect poorer PROs. Additionally, we defined impaired PROs as scores < 40 for survivors’ positive affect and mobility and caregivers’ physical function. For other domains, scores > 60 were considered impaired.

### 2.3. Family Dynamics

The Family Conflict subscale (9 items) of the Family Relationship Index of the Family Environment Scale (FRI-FES) assessed family dynamics [34]. Scores for family conflict were calculated by averaging the scores of the items from this subscale, with higher scores indicating greater family conflict.

### 2.4. County-Level Area Deprivation

Information on area deprivation at the county level was derived from the County Health Rankings & Roadmaps program of the Robert Wood Johnson Foundation [35]. These data encompass information from 3142 counties in the U.S. in 2020, each identified by a 5-digit FIPS code, to capture four domains of deprivation: socioeconomic status or SES (9 indicators), physical environment (5 indicators), health behaviors (9 indicators), and clinical care (7 indicators). The higher scores of each indicator represent the more deprived areas.

To create scores for each area deprivation domain, we first tested bivariate associations of each deprivation indicator with survivors’ PRO domains and selected indicators that were meaningfully and significantly associated with any survivors’ PROs (*p*-values < 0.1). Based on these bivariate associations, we identified the SES and physical environment domains as key area-level determinants of health influencing poor PROs for survivors. Second, we calculated a Z-score for each indicator using indicator-level means and standard deviations from all 3142 counties in the U.S. and classified each county as “deprived” if the Z-score of an indicator was <1 SD (assigned score 1; otherwise, 0). Third, we calculated domain scores for area deprivation domains by summing the scores of all indicators within each domain. The SES domain was scored on a scale from 0 to 7 based on seven indicators (not completing high school, unemployment, children in poverty, children in single-parent households, social association deprivation, violent crime, and injury deaths). The physical environment domain was scored from 0 to 3 based on three indicators (severe housing problems, driving alone to work, and long commute time).

### 2.5. Statistical Analyses

Pearson’s correlation coefficient was used to evaluate the relationship between caregivers’ and survivor’s PROs. Pearson’s correlation coefficient (r) was used to evaluate the relationship between caregivers’ and survivor’s PROs. We set the absolute values 0–0.19 as very weak, 0.2–0.39 as weak, 0.40–0.59 as moderate, 0.6–0.79 as strong, and 0.8–1 as very strong correlations [36]. Spatial analysis techniques were employed to chart the geographic distribution of area deprivation across counties in the U.S. where study participants resided. We calculated bivariate linear regression models to test associations between each determinant (i.e., caregiver’s PROs, parenting behavior, family dynamics, area deprivation) and each PRO domain of survivors. Determinants meeting the criteria for variable selection (*p* < 0.1) were included in multivariable linear regression models, adjusting for survivors’ age, sex, cancer diagnoses, and years from diagnosis. Cohen’s effect size was employed to aid in the interpretation of PRO scores, categorizing effect sizes into small (0.2–0.49 or 2.0–4.9 points on the PROMIS scale), moderate (0.5–0.79 or 5.0–7.9 points on the PROMIS scale), and large (≥0.8 or ≥8.0 points on the PROMIS scale).

All analyses were performed using SAS version 9.3 (SAS Institute, Cary, NC, USA), and spatial patterns of area deprivation were visualized using the ggmap package in R version 4.0.3.

## 3. Results

### 3.1. Participant Characteristics

Table 1 describes the characteristics of study participants. Survivors had a mean (SD) age of 14.2 years (2.9), and the mean time since diagnosis was 10.9 years (2.9). Approximately 50% of survivors were male, 70% were non-Hispanic white, and 50% had been treated for solid tumors, 37% for hematologic cancers, and 14% for central nervous system (CNS) tumors. The mean (SD) age of primary caregivers was 42.2 years (7.4), and over 85% were female. About 64% of survivors were under the coverage of private health insurance, 33% were under Medicaid and or state plans, such as the State Children’s Health Insurance Program, and 4% had no insurance coverage.

### 3.2. Area Deprivation among Study Participants

Figure 1 illustrates the geographical distribution of deprived and non-deprived areas, categorized by SES and physical environment domains, according to the residential counties of study survivors. Appendix A shows the percentage of all U.S. counties designated as deprived status and the percentage of survivors living in these deprived areas. Approximately 61% of survivors lived in a county described as having deprivation in the SES domain, and 44% of survivors lived in a county described as having deprivation in the physical environment domain. Within the SES domain, the most prevalent indicators for the counties where survivors resided were violent crime deprivation (43.2%), children in single-parent households (26.4%), and not completing high school (22.9%). Within the physical environment domain, the most prevalent indicators were a long commute time (18.3%), an environment including risks of severe housing problems (17.6%), and driving alone to work (16.2%).

### 3.3. PROs of Cancer Survivors and Primary Caregivers

Table 2 shows survivors’ and caregivers’ PRO scores and impaired status by individual domains. Among survivors, the most prevalent domains were sleep disturbance (21.2%), followed by psychological stress (19.8%), fatigue (18.4%), depression (17.8%), poor mobility (17.4%), and low positive affect (17.4%). Among primary caregivers, the most prevalent domains were anxiety and poor physical functioning (19.5% on both), followed by depression (17.8%), pain intensity (15.4%), fatigue (14.7%), and sleep disturbance (12.0%).

Table 3 shows correlations for the PRO domains between survivors and caregivers. Caregivers’ anxiety scores were significantly associated with a range of survivors’ PROs in a weak magnitude, including depression (Pearson’s correlation [r] = 0.23, *p* < 0.001), psychological stress (r = 0.22, *p* = 0.0002), fatigue (r = 0.25, *p* < 0.001), and sleep disturbance (r = 0.28, *p* < 0.001). Similar associations were found between caregivers’ depression and a range of survivors’ PROs. Additionally, caregivers’ sleep disturbance was significantly associated with survivors’ fatigue (r = 0.20, *p* = 0.001) and poor mobility (r = −0.20, *p* = 0.001).

### 3.4. Associations between Multilevel Social Determinants and Poor PROs of Cancer Survivors

Table 4 shows the multivariable associations between social determinants (i.e., caregivers’ PROs, parenting behavior, family dynamics, and area deprivation selected from the bivariate analysis) and the individual PRO domains of the survivors (see Appendix A for the bivariate associations between each determinant and each PRO domain). Female survivors reported higher depression (B = 3.64, 95% CI = 1.63, 5.65) and psychological stress (B = 4.94, 95% CI = 2.82, 7.07) scores compared with male survivors, representing a small effect size. Survivors who had been diagnosed with CNS tumors had greater fatigue (B = 4.76, 95% CI = 0.78, 8.73) and lower mobility (B = −4.94, 95% CI = −7.61, −2.28) scores compared to survivors who had been diagnosed with hematologic cancers, representing a small effect size.

At the caregiver/family level, higher anxiety scores in caregivers were significantly associated with higher scores in survivors for the depression (B = 0.17, 95% CI = 0.05, 0.30), psychological stress (B = 0.20, 95% CI = 0.07, 0.33), fatigue (B = 0.29, 95% CI = 0.13, 0.45), and sleep disturbance (B = 0.26, 95% CI = 0.12, 0.39) domains and lower scores in the positive affect domain (B = −0.13, 95% CI = −0.24, −0.02). Higher sleep disturbance scores in caregivers were significantly associated with lower mobility scores in survivors (B = −0.13, 95% CI = 0.25, −0.003). Therefore, caregivers’ PROs, when differing by 25 points on the PROMIS scale (i.e., comparing individuals from one quarter to those from the adjacent quarter), were associated with a variation in survivors’ PROs, indicating a small to moderate effect size. For instance, a comparison of fatigue reported by caregivers between one quarter and the adjacent quarter revealed a moderate effect size difference in psychological stress among survivors from these quarters, with a coefficient of B = 7.25, or equivalently, B = 0.29 times 25 points. Higher family conflict scores were significantly associated with higher sleep disturbance in survivors (B = 0.13, 95% CI = 0.01, 0.25).

At the neighborhood level, survivors residing in areas with a greater deprivation in SES had more sleep disturbance (B = 0.94, 95% CI = 0.11, 1.78), and those residing in areas with a greater deprivation in the physical environment had more psychological stress (B = 2.26, 95% CI = 0.61, 3.91) and fatigue (B = 2.05, 95% CI = 0.09, 4.01), as well as lower positive affect (B = −1.89, 95% CI = −3.44, −0.34) and lower mobility (B = −1.51, 95% CI = −2.81, −0.20). Therefore, survivors residing in areas classified as having a deprived physical environment experienced worse psychological stress and fatigue scores, with a moderate effect size, compared to those living in areas not classified as deprived.

## 4. Discussion

This study identified that the well-being of young survivors of childhood cancer represented by a broad spectrum of PROs is significantly associated with a broad spectrum of social factors at the caregiver, family, and neighborhood levels. Notably, caregivers’ self-reported poorer anxiety and sleep disturbance, family conflict, and higher neighborhood-level deprivation emerged as pivotal correlates with poorer PROs in survivors. These findings highlight the importance of considering not only survivor factors (e.g., diagnosis, treatment, and age at diagnosis) but also the mental health of caregivers and the environment where survivors live. A comprehensive approach that includes an assessment of family and neighborhood factors as employed in our study represents a significant and innovative advance that can provide important perspectives to guide survivorship research and clinical care.

Extant evidence suggests that elevated anxiety, depression, psychological stress, and post-traumatic stress symptoms in caregivers place young pediatric cancer survivors at risk of stressful living circumstances and poor HRQOL [15,17,18,37]. In contrast, lower psychosocial family risk and lower levels of parental psychological distress are associated with better HRQOL for children in the first year following a cancer diagnosis [38]. However, the impact of these factors as survivors of childhood cancer age through the critical period of adolescence has not, to our knowledge, been described. Extending from previous studies, our study indicates that both family conflict and caregivers’ PROs (most commonly anxiety and sleep disturbance) are significantly associated with poorer PROs in survivors. Specifically, higher anxiety in caregivers is associated with both psychological (i.e., depression, psychological stress, low positive affect) and physical (i.e., fatigue, sleep disturbance) PROs in young children several years after their cancer treatment has ended. Caregiver psychological status may likely influence a survivor’s PROs through bio-psycho-social mechanisms by affecting the parent’s ability to provide needed emotional support for the child’s adjustment, increasing the parent’s use of behaviors associated with child distress (e.g., hostility, withdrawal), and making the child more susceptible to psychological disorders (e.g., depression, anxiety, post-traumatic stress) [39]. Positive aspects of the family context may be protective against the negative effects of parental distress on the HRQOL of pediatric cancer survivors; therefore, assessing parental distress and overall family functioning should be an integrated part of routine follow-up visits for pediatric cancer survivorship care [30].

Population-based, large-scale studies among adult patients in the general population with chronic health conditions or survivors of adult-onset cancers show poorer self-rated health and HRQOL among those living in deprived areas compared to those living in non-deprived areas [21,40], though the evidence in survivors of childhood cancer is limited. In this study, we found that living in counties deemed as deprived due to SES and physical environmental circumstances is linked to psychological stress, fatigue, and sleep disturbances, as well as diminished positive affect and mobility among young cancer survivors. This discovery broadens previous investigations into the social determinants of health in childhood cancer, where the focus has predominantly been on survival or mortality rather than on survivor quality of life and well-being measured by PROs [25,41,42]. One possible reason underlying the association between area deprivation and poor PROs is that cancer survivors living in disadvantaged neighborhoods are less likely to access community-level resources and support to address survivor-related healthcare needs [25]. If survivors are continuously exposed to chronic stress from their environment (e.g., crowded housing conditions, high crime rate, untrustworthy neighborhood) and have limited community infrastructures (e.g., fitness or recreational facilities, food desert/lack of nutritious foods) [43], the inadequate neighborhood environment may influence the sustainability of health behaviors and adherence to follow-up care, which may lead to an elevated risk of late effects and poor PROs [21].

This study identified significant family-level (e.g., family conflict) and neighborhood-level (e.g., socioeconomic and physical environment) factors that were associated with poor PROs among childhood cancer survivors. Evaluating family- and neighborhood-level determinants regularly for childhood cancer survivors and targeting specific factors for each cancer survivor is crucial for developing comprehensive and individually tailored healthcare strategies for young childhood cancer survivors. Tailored interventions that consider the unique social contexts of each child can contribute to improved PROs. Families with psychosocial family risk may need other support, such as social work or psychological services, to strengthen psychosocial survivorship care. Family-level interventions, such as the FAMily-Oriented Support (FAMOS) family therapy program, have been shown to reduce caregiver, especially maternal, depression and in turn improve psychological reactions in young cancer patients [44]. While monitoring neighborhood deprivation requires resources beyond the scope of clinical care, clinicians may intervene on social determinants negatively affecting health by referring young survivors and their caregivers to psychologists, social workers, or community-based resources that address problems such as financial hardship, housing, and transportation issues [45,46]. It is also crucial to acknowledge that addressing environmental factors (such as the high rates of violent crime, unemployment, and severe housing conditions highlighted in our study) poses a greater challenge in the short term compared to family factors. Addressing neighborhood-related issues requires a coordinated effort across multiple levels, from community organizations to governmental agencies, to tackle a range of interconnected factors, including occupation and job status, income levels, educational opportunities, transportation, housing conditions, crime rates, community safety, and racial segregation.

This study has several limitations. First, our data were collected from pediatric cancer survivors in a single pediatric cancer center and resided in the southeastern U.S., which limits the generalizability of our findings to all young pediatric cancer survivors in the U.S. Second, our study is based on a cross-sectional design, and social determinants of PROs were assessed at the time of the study. These determinants likely identified at the cancer diagnosis and the change from cancer diagnosis to later time points would also be associated with poorer PROs. Future longitudinal studies are needed to identify the causal nature of the associations identified in this study. Finally, this study only focused on poor PROs as the outcome of interest. Future studies are encouraged to test associations between multilevel social determinants and healthcare use and clinically ascertained outcomes (e.g., unexpected healthcare utilization, the onset of chronic health conditions, neurocognitive/physical performance deficits, premature death).

## 5. Conclusions

Multilevel social determinants, especially caregivers’ poorer PROs and county-level adversity in socioeconomic status and physical environment, contributed to the poor PROs of pediatric cancer survivors. These results highlight the significance of assessing survivors for challenges related to multilevel determinants of health throughout their cancer journey. Given that clinical guidelines recommend yearly psychosocial screening for survivors of pediatric cancer and psychosocial follow-up as a standard of care in pediatric psychosocial oncology [24,47], it is crucial to include evaluations of family- and neighborhood-level determinants, alongside caregiver and pediatric PROs, when caring for childhood cancer survivors.

## Figures and Tables

**Figure 1 cancers-16-01661-f001:**
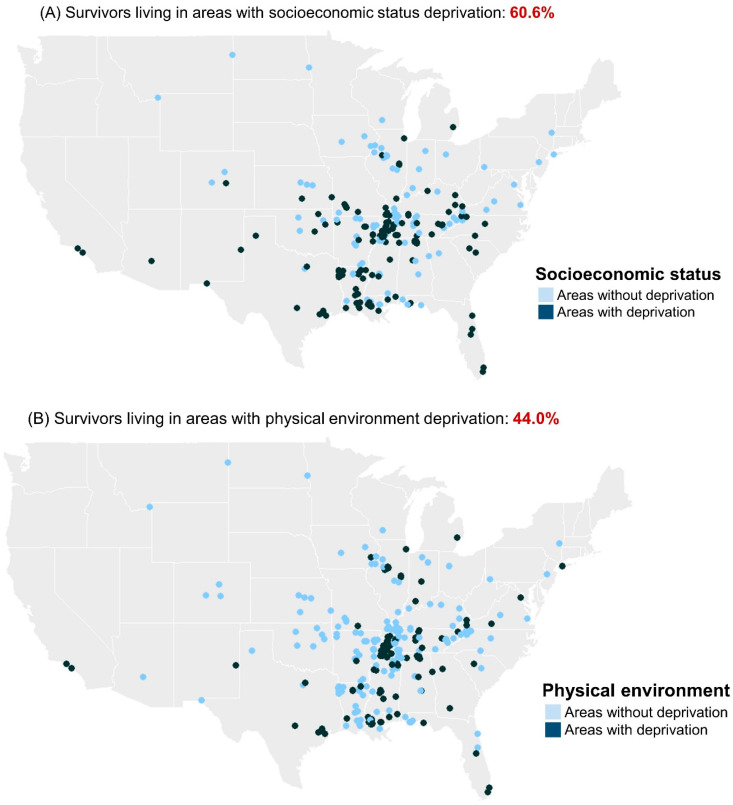
Distribution of deprived and non-deprived areas where young survivors of childhood cancer reside, categorized by socioeconomic status and physical environment domains.

**Table 1 cancers-16-01661-t001:** Demographic and treatment characteristics and caregiver and family factors of survivors of childhood cancer.

Characteristics	Survivors (*n* = 293)	
Survivor factors		
Age at study in years (mean, SD)	14.2	2.9
Time since cancer diagnosis in years (mean, SD)	10.9	2.9
Sex (*n*, %)		
Male	147	50.2
Female	146	49.8
Race/ethnicity (*n*, %)		
White, non-Hispanic	202	69.9
Black, non-Hispanic	52	18.0
Hispanic	30	10.4
Other	5	1.7
Health insurance coverage (*n*, %)		
Private	183	63.5
Public (Medicaid or other state insurance)	94	32.6
Child not insured	11	3.8
Pediatric cancer diagnosis (*n*, %)		
Leukemia	96	32.8
Lymphoma	13	4.4
Central nervous system (CNS) tumors	41	14.0
Solid tumors	143	48.8
Cancer treatment (*n*, %)		
Any radiotherapy	82	28.0
Any chemotherapy	235	80.2
Anthracyclines	161	55.0
Classic alkylating agents	170	58.0
Corticosteroids	91	31.1
High-dose cytarabine	51	17.4
High-dose methotrexate	87	29.7
Epipodophyllotoxins	88	30.0
Vincristine	188	64.2
Any invasive surgery	211	72.0
	Primary Caregivers (*n* = 293)	
Caregiver and family factors		
Age at study in years (mean, SD)	42.2	7.4
Number of self-reported chronic diseases (mean, SD)	1.0	1.3
Sex (*n*, %)		
Male	42	14.5
Female	247	85.5
Race/ethnicity (*n*, %)		
White, non-Hispanic	212	73.4
Black, non-Hispanic	52	18.0
Hispanic	22	7.6
Other	3	1.0
Marital status (*n*, %)		
Single	213	74.5
Married/living with a partner	23	8.0
Widowed/divorced/separated	50	17.5
Mother’s education (*n*, %)		
HS graduate/GED or below	147	50.5
Some college/training after HS	88	30.2
College graduate/post-graduate	56	19.2
Annual household income (*n*, %)		
<USD 35,000	73	24.9
USD 35,000–USD 74,999	98	33.5
≥USD 75,000	122	41.6
Parenting behavior (mean, SD)		
Overprotection	24.6	6.4
Vulnerability	3.3	2.8
Family dynamics (mean, SD)		
Cohesion	56.2	10.7
Conflict	43.9	9.0
Expressiveness	55.7	10.7

Abbreviations: SD, standard deviation; HS, high school; GED, General Educational Development certificate.

**Table 2 cancers-16-01661-t002:** Distributions of individual PRO domain scores and PRO impairment status among survivors and caregivers.

PRO Domains	Mean	SD	% with Impairment *
Survivors			
Depression ^†^	44.0	8.8	17.8
Psychological stress ^†^	48.8	9.7	19.8
Fatigue ^†^	40.9	11.3	18.4
Sleep disturbance ^†^	50.7	9.3	21.2
Positive affect ^‡^	53.0	8.8	17.4
Mobility ^‡^	53.3	7.4	17.4
Caregivers			
Anxiety ^†^	48.7	9.1	19.5
Depression ^†^	44.7	7.2	17.8
Fatigue ^†^	46.7	8.7	14.7
Sleep disturbance ^†^	49.9	7.9	12.0
Pain intensity ^†^	47.1	8.3	15.4
Physical function ^‡^	55.2	6.5	19.5

Abbreviations: PROs, patient-reported outcomes; SD, standard deviation. * Impairment in each PRO domain is defined as a score <40 for survivors’ positive affect and mobility and caregivers’ physical function. For all other domains, impaired is defined as a score > 60. ^†^ Higher score indicates poorer PROs. ^‡^ Higher score indicates better PROs.

**Table 3 cancers-16-01661-t003:** Correlations of PROs between cancer survivors and primary caregivers.

PROs of Caregivers	PROs of Survivors					
	Depression ^†^	Psychological Stress ^†^	Fatigue ^†^	Sleep Disturbance ^†^	Positive Affect ^‡^	Mobility ^‡^
	r (*p*-value)	r (*p*-value)	r (*p*-value)	r (*p*-value)	r (*p*-value)	r (*p*-value)
Anxiety ^†^	0.23 (<0.001)	0.22 (0.0002)	0.25 (<0.001)	0.28 (<0.001)	−0.16 (0.01)	−0.14 (0.02)
Depression ^†^	0.27 (<0.0001)	0.23 (<0.0001)	0.24 (<0.0001)	0.19 (0.001)	−0.15 (0.01)	−0.17 (0.004)
Fatigue ^†^	0.17 (0.01)	0.21 (0.0003)	0.25 (<0.001)	0.22 (<0.001)	−0.13 (0.03)	−0.18 (0.003)
Sleep disturbance ^†^	0.15 (0.01)	0.14 (0.02)	0.20 (0.001)	0.19 (0.001)	−0.06 (0.34)	−0.20 (0.001)
Pain intensity ^†^	0.08 (0.19)	−0.02 (0.78)	0.09 (0.14)	0.07 (0.27)	−0.01 (0.92)	−0.07 (0.24)
Physical function ^‡^	−0.03 (0.57)	−0.03 (0.65)	−0.10 (0.08)	−0.05 (0.41)	0.05 (0.39)	0.08 (0.19)

Abbreviations: PROs, patient-reported outcomes; r, Pearson’s correlation coefficient. ^†^ Higher score indicates poorer PROs. ^‡^ Higher score indicates better PROs.

**Table 4 cancers-16-01661-t004:** Multivariable associations of contextual/social determinants with PROs among young survivors of childhood cancer.

Factors	Depression ^†^	Psychological Stress ^†^	Fatigue ^†^	Sleep Disturbance ^†^	Positive Affect ^‡^	Mobility ^‡^
	B (95% CI)	B (95% CI)	B (95% CI)	B (95% CI)	B (95% CI)	B (95% CI)
Survivor factors (personal level)						
Age	-	0.37 (−0.11, 0.85)	-	0.38 (0.01, 0.76)	−0.49 (−0.84, −0.13) **	0.28 (−0.02, 0.58)
Time since cancer diagnosis	-	0.22 (−0.25, 0.69)	-	-	-	-
Sex						
Male	Ref	Ref	Ref	-	-	-
Female	3.64 (1.63, 5.65) ***	4.94 (2.82, 7.07) ***	2.06 (−0.47, 4.60)	-	-	-
Cancer diagnosis						
Hematologic cancers	-	-	Ref		-	Ref
CNS tumors			4.76 (0.78, 8.73) **	-	-	−4.94 (−7.61, −2.28) ***
Solid tumors			−1.01 (−3.79, 1.77)	-	-	−0.46 (−2.34, 1.41)
Caregiver and family factors (family level)						
Annual household income						
≥USD 75,000	Ref	-	-	-	-	-
<USD 75,000	1.08 (−0.99, 3.14)	-	-	-	-	-
# of caregiver chronic diseases	0.46 (−0.34, 1.26)	0.44 (−0.40, 1.28)	0.41 (−0.57, 1.39)	-	-	0.003 (−0.67, 0.67)
Family dynamics						
Conflict	0.03 (−0.09, 0.15)	0.06 (0.07, 0.18)	-	0.13 (0.01, 0.25) *	−0.11 (−0.22, 0.003)	−0.07 (−0.17, 0.02)
Caregiver PROs						
Anxiety	0.17 (0.05, 0.30) **	0.20 (0.07, 0.33) **	0.29 (0.13, 0.45) ***	0.26 (0.12, 0.39) ***	−0.13 (−0.24, −0.02) *	−0.07 (−0.17, 0.04)
Sleep disturbance	0.05 (−0.09, 1.96)	0.01 (−0.14, 0.17)	0.14 (0.05, 0.32)	0.06 (−0.09, 0.21)	-	−0.13 (0.25, −0.003) *
Neighborhood factors (county level)						
Socioeconomic status	-	-	-	0.94 (0.11, 1.78) *	-	-
Physical environment	-	2.26 (0.61, 3.91) **	2.05 (0.09, 4.01) *	1.27 (−0.44, 2.98)	−1.89 (−3.44, −0.34) *	−1.51 (−2.81, −0.20) *

Abbreviation: PROs, patient-reported outcomes; B, regression coefficient; CI, confidence interval; Ref, reference group. ** p* < 0.05, *** p* < 0.01, **** p* < 0.001. ^†^ Higher score indicates poorer PROs. ^‡^ Higher score indicates better PROs.

## Data Availability

The data can be shared up on request.

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
