# Peer review of "Multilevel Social Determinants of Patient-Reported Outcomes in Young Survivors of Childhood Cancer"

_cancers, 2024, doi:10.3390/cancers16091661_

Round 1
Reviewer 1 Report
Comments and Suggestions for Authors
The authors explore the physical and psychosocial PROs in cancer survivors under the age of 18. This is a step beyond previous research as reviewed in the paper by Barnett et al. in the J.Cancer Surviv.
The sample size and the observed statistics compel me to see the strength in the work performed.
Further, highlights the importance of social determinants on survival in pediatric cancers.
1. It may be important to include more information about treatments - (number of in-patients and outpatient visits). The type of insurance(private vs government) is yet another important socioeconomic determinant as it may allude to the financial burden of the treatment.
2. Condensing the statements made to slightly improve clarity is suggested.
Comments on the Quality of English Language
Condensing the language of the article may improve clarity and make the statements made in the work more direct.
Author Response
REVIEWER 1 COMMENTS:
The authors explore the physical and psychosocial PROs in cancer survivors under the age of 18. This is a step beyond previous research as reviewed in the paper by Barnett et al. in the J. Cancer Surviv. The sample size and the observed statistics compel me to see the strength in the work performed. Further, highlights the importance of social determinants on survival in pediatric cancers.
1. It may be important to include more information about treatments - (number of in-patient and outpatient visits). The type of insurance (private vs government) is yet another important socioeconomic determinant as it may allude to the financial burden of the treatment.
Responses:
Thank you for your useful comment. This cross-sectional study included 293 dyads of childhood cancer survivors and their primary caregivers who visited St. Jude Children’s Research Hospital (SJCRH) between July 2017 and June 2018 for annual follow-up; therefore, the research was conducted in an outpatient setting. To clarify the “treatments” factors, we added specific types of cancer treatments to the revised Table 1. For insurance coverage, we have already reported this information in Table 1. In the revised manuscript, we further added a statement under the header “Participant characteristics” of the Result section, which now reads: “About 64% of survivors were under the coverage of private health insurance, 33% were under Medicaid and or state plans, such as the State Children's Health Insurance Program, and 4% had no insurance coverage.”
2. Condensing the statements made to slightly improve clarity is suggested.
Responses:
Thank you. We have revised the manuscript by condensing some statements to enhance clarity and flow.
Reviewer 2 Report
Comments and Suggestions for Authors
This interesting work deals with a current, critical, and delicate aspect of the difficult path of pediatric oncology treatment.
Exploration of physical and emotional risk factors from both the patient and caregiver perspectives, as presented in this work, is crucial for enhancing the quality of life in pediatric oncology treatment.
In this paper, I would like to underline some points to clarify/improve:
- In the statistical analysis paragraph, it would be beneficial if the authors could provide more detailed information on the geocoding process and specify the acceptable value(s) for the Pearson coefficient. This would further enhance the credibility of their work.
- For ease of reading and chronological order of citation, I suggest moving figure 1 before table 2.
- The authors conclude that evaluating the family and neighbourhood alongside the surviving children and caregivers is fundamental. How could these suggestions be put to use? What aspects could be improved on an environmental level, and what additional support could be offered to families?
Author Response
REVIEWER 2 COMMENTS:
This interesting work deals with a current, critical, and delicate aspect of the difficult path of pediatric oncology treatment. Exploration of physical and emotional risk factors from both the patient and caregiver perspectives, as presented in this work, is crucial for enhancing the quality of life in pediatric oncology treatment.
In this paper, I would like to underline some points to clarify/improve:
1. In the statistical analysis paragraph, it would be beneficial if the authors could provide more detailed information on the geocoding process and specify the acceptable value(s) for the Pearson coefficient. This would further enhance the credibility of their work.
Responses:
Thank you for this well-taken point. Per the reviewer’s suggestion, we complemented information about the geocoding process under the header “Study participants and data collection” of the Method section, which now reads: “We used a geocoding approach to obtain the Federal Information Processing Standards (FIPS) codes based on the home address information of each participant, including the structure number, street name, city, state, and zip code. For this purpose, we utilized the Geocoding Service API provided by the U.S. Census Bureau (https://geocoding.geo.census.gov/geocoder/) to transform each address into geographical coordinates, specifically latitude and longitude. Using the latitude and longitude data, we linked the address of each study participant to FIPS codes, which provide location-specific information in the U.S., including demographic, environmental, and administrative datasets.”
Regarding the thresholds for Pearson’s correlation coefficients (r), we consider the absolute value 0-0.19 as very weak, 0.2-0.39 as weak, 0.40-0.59 as moderate, 0.6-0.79 as strong and 0.8-1 as very strong correlations. In the revised manuscript, we complemented this information under the header “Statistical Analyses” of the Method section, which now reads: “Pearson's correlation coefficient (r) was used to evaluate the relationship between caregivers’ and survivor’s PROs. We set the absolute value 0-0.19 as very weak, 0.2-0.39 as weak, 0.40-0.59 as moderate, 0.6-0.79 as strong, and 0.8-1 as very strong correlations [36].”
[36] Reference: Correlation and regression. Statistics at Square One, Campbell, M.J., Ed.; 2021; pp. 165-182
2. For ease of reading and chronological order of citation, I suggest moving figure 1 before table 2.
Responses:
Per this reviewer’s suggestion, we have moved Figure 1 before Table 2.
3. The authors conclude that evaluating the family and neighbourhood alongside the surviving children and caregivers is fundamental. How could these suggestions be put to use? What aspects could be improved on an environmental level, and what additional support could be offered to families?
Responses:
Thanks for this useful comment. This study identified significant family-level (e.g., family conflict) and neighborhood-level (e.g., socioeconomic and physical environment) factors that were associated with poor PROs in childhood cancer survivors. It is important to provide interventions to improve these contextual factors of poor PROs. In the revised manuscript, we suggest several strategies targeting family factors and environmental factors. However, it's important to recognize that addressing environmental factors (such as high rates of violent crime, unemployment, and severe housing conditions as reported in our study) presents a greater challenge compared to family factors. These issues demand a concerted effort that spans multiple levels, from the community to governmental agencies.
We have clarified our points in the 4th paragraph of the Discussion section, which now reads: “This study identified significant family-level (e.g., family conflict) and neighborhood-level (e.g., socioeconomic and physical environment) factors that were associated with poor PROs among childhood cancer survivors. Evaluating family and neighborhood-level determinants regularly for childhood cancer survivors and targeting specific factors for each cancer survivor is crucial for developing comprehensive and individually-tailored healthcare strategies for young childhood cancer survivors. Tailored interventions that consider the unique social contexts of each child can contribute to improved PROs. Families with psychosocial family risk may need other support, such as social work or psychological services, to strengthen psychosocial survivorship care. Family-level interventions, such as the FAMily-Oriented Support (FAMOS) family therapy program, have been shown to reduce caregiver, especially maternal, depression and in turn improve psychological reactions in young cancer patients [44]. While monitoring neighborhood deprivation requires resources beyond the scope of clinical care, clinicians may intervene on social determinants negatively affecting health by referring young survivors and their caregivers to psychologists, social workers, or community-based resources that address problems such as financial hardship, housing, and transportation issues [45,46]. It's also crucial to acknowledge that addressing environmental factors (such as the high rates of violent crime, unemployment, and severe housing conditions highlighted in our study) poses a greater challenge in the short term compared to family factors. Addressing neighborhood-related issues requires a coordinated effort across multiple levels, from community organizations to governmental agencies, to tackle a range of interconnected factors, including occupation and job status, income levels, educational opportunities, transportation, housing conditions, crime rates, community safety, and racial segregation.”